# Transcranial focused ultrasound-mediated neurochemical and functional connectivity changes in deep cortical regions in humans

Siti N. Yaakub [1,2], Tristan A. White[1,2], Jamie Roberts [3], Eleanor Martin [4,5], Lennart Verhagen [6], Charlotte J. Stagg [7,8], Stephen Hall[1,2] & Elsa F. Fouragnan [1,2] ✉

Low-intensity transcranial ultrasound stimulation (TUS) is an emerging non-invasive technique for focally modulating human brain function. The mechanisms and neurochemical substrates underlying TUS neuromodulation in humans and how these relate to excitation and inhibition are still poorly understood. In 24 healthy controls, we separately stimulated two deep cortical regions and investigated the effects of theta-burst TUS, a protocol shown to increase corticospinal excitability, on the inhibitory neurotransmitter gamma-aminobutyric acid (GABA) and functional connectivity. We show that theta-burst TUS in humans selectively reduces GABA levels in the posterior cingulate, but not the dorsal anterior cingulate cortex. Functional connectivity increased following TUS in both regions. Our findings suggest that TUS changes overall excitability by reducing GABAergic inhibition and that changes in TUS-mediated neuroplasticity last at least 50 mins after stimulation. The difference in TUS effects on the posterior and anterior cingulate could suggest state- or location-dependency of the TUS effect—both mechanisms increasingly recognized to influence the brain's response to neuromodulation.

Low-intensity focused transcranial ultrasound stimulation (TUS) is a non-invasive neuromodulation technique that has shown promise in a range of applications from basic neuroscience research to therapeutic applications in neurological and psychiatric diseases. Compared with other non-invasive neuromodulatory techniques such as transcranial magnetic stimulation (TMS) and transcranial direct current stimulation (tDCS), TUS can target both cortical and deep brain regions with very high spatial specificity (in the order of millimetres vs centimetres in TMS and tDCS)[1]. Depending on the sonication paradigm used, the neuromodulatory effects of TUS can be limited to the period during or immediately after stimulation ("online" effects), or can last several minutes to hours after stimulation ("offline" effects)[1]. Offline TUS effects are of particular interest because they may reflect long-term potentiation/depression-like neuroplasticity[2], lasting longer than transient neuronal adaption effects, with the potential to be used to modulate aberrant activity in brain regions or networks for therapeutic applications. It is thought that TUS induces neuromodulation primarily through mechanical interactions of the ultrasound wave as it passes through cells at the target location[3,4]. However, the mechanism by which this translates into excitatory or inhibitory neuromodulation and its effects on large-scale human brain connectivity remains unclear.

[1]School of Psychology, Faculty of Health, University of Plymouth, Plymouth, UK. [2]Brain Research and Imaging Centre, Faculty of Health, University of Plymouth, Plymouth, UK. [3]Department of Clinical Measurement and Innovation, University Hospitals Plymouth NHS Trust, Plymouth, UK. [4]Department of Medical Physics and Biomedical Engineering, University College London, London, UK. [5]Wellcome/EPSRC Centre for Interventional and Surgical Sciences, University College London, London, UK. [6]Donders Institute for Brain, Cognition and Behaviour, Radboud University Nijmegen, Nijmegen, Netherlands. [7]Wellcome Centre for Integrative Neuroimaging, FMRIB, Nuffield Department of Clinical Neurosciences, University of Oxford, Oxford, UK. [8]MRC Brain Network Dynamics Unit, University of Oxford, Oxford, UK. ✉e-mail: elsa.fouragnan@plymouth.ac.uk

The combination of offline TUS with the high spatial resolution of magnetic resonance imaging (MRI) allows the measurement of TUS effects at both the local level, in individual target regions, and at the network level across the whole brain, including in deep brain regions. Previous studies in both macaques and humans have used functional magnetic resonance imaging (fMRI) and arterial spin labelling to show large-scale changes in brain activity and perfusion due to TUS. In macaques, TUS of deep cortical and sub-cortical regions have shown changes in task-based fMRI[5] and behaviour[6] and in resting-state fMRI (rsfMRI) connectivity profiles of targeted regions[7,8]. In humans, TUS has been shown to affect changes in both rsfMRI connectivity and regional perfusion[9,10].

With magnetic resonance spectroscopy (MRS), it is possible to quantify in vivo levels of gamma-aminobutyric acid (GABA), the major inhibitory neurotransmitter and glutamate, the main excitatory neurotransmitter, providing insights into GABAergic and glutamatergic mechanisms of TUS-induced neuroplasticity. MRS measures of GABA are unable to distinguish between intra- or extracellular GABA, but are thought to represent tonic inhibition and the overall inhibitory "tone" of the region[11], rather than phasic or synaptic inhibition[12,13]. In rats, TUS has been shown to reduce extracellular GABA with no change in glutamate levels up to 120 mins after intervention[14]. To date, MRS has not been exploited to explore the neurochemical basis of TUS neuromodulation in humans.

Here, we investigate whether TUS can induce offline changes in two deep cortical regions with well-defined and separable connectivity profiles at rest: the dorsal anterior cingulate cortex (dACC), part of the salience network[15] and the posterior cingulate cortex (PCC), a major hub of the default mode network, which is most active during wakeful rest[16]. Aberrant functional connectivity in these networks has been implicated in several neurological and psychiatric disorders[17], making these regions potential targets for therapeutic TUS applications.

Using MRS and rsfMRI with a theta-burst TUS protocol shown to induce offline increases in corticospinal excitability[18], we stimulated each region in separate sessions and compared the effects with a sham stimulation. We show a selective reduction in GABA levels only in the PCC at ~20–30 mins post-stimulation. We also show increased functional connectivity following TUS of both regions, with the greatest increases in rsfMRI connectivity occurring at later (~46 mins post-TUS) compared to earlier (~13 mins post-TUS) time points. Additionally, stimulation of the PCC also increased functional connectivity of the dACC, but not vice versa. Importantly, we also show, via acoustic measurements and simulations, that we were able to effectively and safely target both deep cortical regions in all individuals in our study, something that is often overlooked in TUS studies of deeper brain regions. Our results demonstrate in vivo GABA changes modulated by TUS, and functional connectivity changes evolving over time and lasting at least 50-mins post-stimulation. The disparate findings between PCC and dACC stimulation suggest a possible state- or location-dependence of TUS, with important implications for the design and development of future TUS research in humans.

## Results

In this preregistered study (https://osf.io/bcf4v) of 24 healthy adults, we investigated regionally specific TUS-induced changes in GABA, glutamate and functional connectivity, by comparing MRS and rsfMRI following TUS of left ACC, TUS of left PCC, or sham TUS (see study design in Fig. 1). Acoustic simulations were performed on a subset of participants ($n = 4$) after Session 1 (the MRI-only session) and before the three TUS & MRI sessions for each TUS target location to ensure we remained within TUS safety guidelines (for further details, see methods). Acoustic simulations for the remaining participants were performed at the end of the study. All participants completed three TUS and MRI sessions, in which TUS was applied either to the left dACC or left PCC, or sham TUS (no stimulation), followed by MRI scans. The

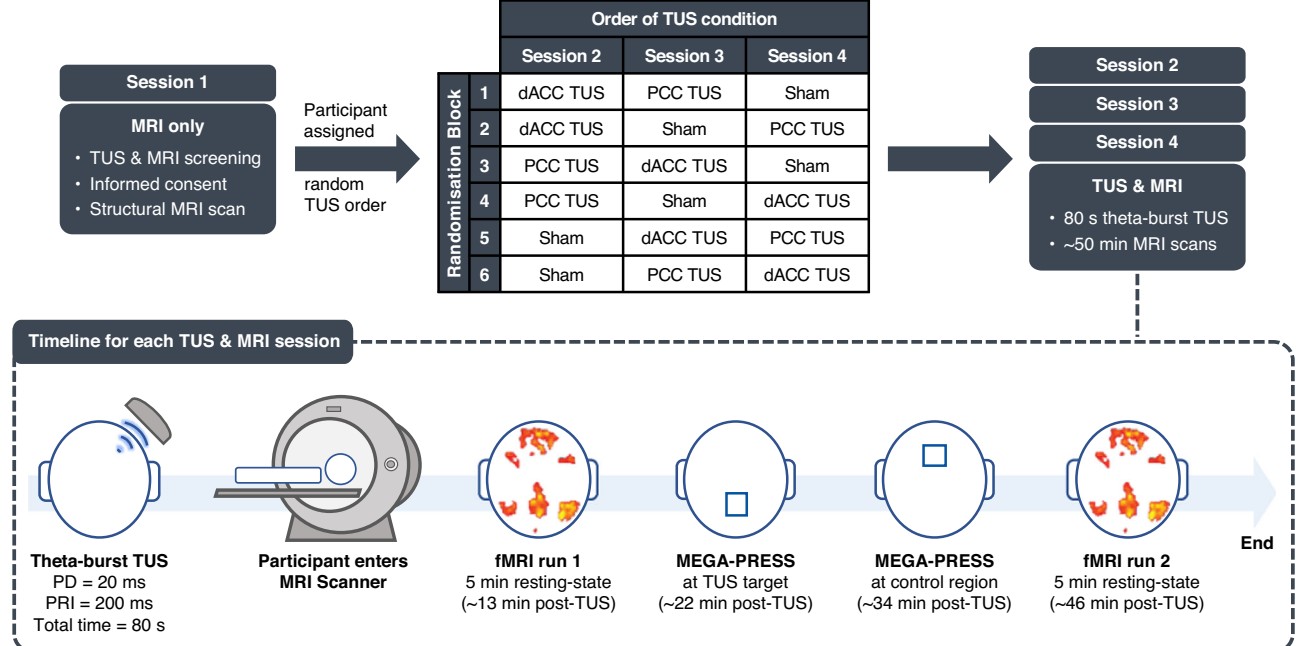

**Fig. 1 | Study design.** Participants ($n = 24$) first attended an MRI-only session where they had a structural MRI scan and were assigned to one of six randomisation blocks, which determined the order of the transcranial ultrasound stimulation (TUS) conditions (counterbalanced across participants). The structural MRI was used to plan and target TUS for the subsequent three study sessions involving either sham TUS or TUS applied to the left dorsal anterior cingulate cortex (dACC) or left posterior cingulate cortex (PCC), immediately followed by a series of MRI scans. Scans included a 5-minute resting-state fMRI run, MEGA-PRESS MRS acquired at the TUS target region, MEGA-PRESS MRS acquired at the control region (i.e., region not targeted with TUS during that session), and another 5-minute resting-state fMRI run. Sessions took place at approximately the same time of day for each participant, and at least one week apart. Post-TUS timings shown are from the average across all participants. PD pulse duration, PRI pulse repetition interval.

first rsfMRI run was acquired at $13.1 \pm 2.0$ mins post-TUS. MRS was acquired at the TUS target at $22.3 \pm 2.0$ mins and in the control region (region not targeted with TUS during that session), at $33.7 \pm 2.2$ mins post-TUS. The second rsfMRI scan was acquired at $46.0 \pm 2.3$ mins post-TUS.

We first describe the results of our acoustic simulations and show the simulated transcranial pressure field for each targeted region. Next, we report TUS-mediated changes in GABA and glutamate in the left dACC and left PCC, followed by TUS-mediated changes in functional connectivity of the left dACC and left PCC. Lastly, we describe exploratory analyses of associations between spectroscopy and functional connectivity changes, and changes related to inter-individual differences in simulated transcranial acoustic measures.

## Characterising the ultrasound field with free-field simulations

We simulated the acoustic pressure distribution generated by the transducer in three dimensions in a free field for two focal depths: 60 mm (Fig. 2a) and 69 mm (Fig. 2b), representing the average focal depths across individuals for the left dACC and left PCC regions respectively. At the target spatial-peak pulse-average intensity ($I_{SPPA}$) of 33.8 W/cm$^2$, the maximum pressure at the TUS focus was 1.01 MPa, mechanical index (MI) 1.4 and spatial-peak temporal-average intensity ($I_{SPTA}$) 3380 mW/cm$^2$ before transcranial transmission. Our simulations showed that the length of the focal region along the trajectory of the beam was shorter at 60 mm than at 69 mm (the full width at half maximum, FWHM, along the trajectory was 32.1 mm and 39.4 mm, respectively). The FWHM of the lateral cross-sections of the beam was 4.5 mm at 60 mm and 5.0 mm at 69 mm focal depth, indicating a slightly wider TUS focus at deeper focal depths.

## Transcranial acoustic simulations

We estimated each participant's skull from pseudo computed tomography (CT) images derived from T1-weighted MR images (Fig. 3a). Transcranial simulations showed that the intensity profile remained elliptical, with a similar size and shape to free-field simulations, the trajectory was linear and remained approximately perpendicular to the transducer face which allowed us to reliably target the dACC and PCC (Fig. 3b) in all participants. Transcranial attenuation of focal intensity

was ~58% on average, in line with typical values for attenuation through the skull (c.f. ~51.7% attenuation of intensity in acoustic tank measurement through a section of a skull[19]). The transcranial $I_{SPPA}$ and MI were below the United States Food & Drug Administration (US FDA) recommended limits for both regions. We simulated temperature rise in the two participants with the highest attenuation. The maximum temperature rise was found in the skull below the transducer (1.48 °C and 1.88 °C) and did not exceed 2 °C for either individual.

Table 1 summarises the acoustic properties at the focus for both regions. The parameters used in the simulations and the full results of the acoustic simulations for all study participants are given in Supplementary Tables 1 and 2. Both the dACC and PCC showed similar maximum intensity and pressure at the focus, however, the focal volumes (−6 dB volume, or pressure at FWHM), and hence the volumes overlapping with the MRS voxel, were smaller for dACC simulations than PCC simulations. This is likely because of the elongation of the ultrasound beam seen at deeper focal depths (Fig. 2). Our simulations showed that the elliptical TUS focus largely overlapped with the $2 \times 2 \times 2$ cm$^3$ MRS voxel during each session (Fig. 3c), suggesting consistency in the manual placement of MRS voxels across sessions, and that we were able to stimulate and measure in the same area. We also found that focal volumes were negatively correlated with $I_{SPPA}$ (Pearson's $r = -0.63$, $p = 1.31 \times 10^{-6}$), such that the larger the focal volume, the lower the $I_{SPPA}$.

## Side effects associated with TUS

The day after each study session, participants were sent a TUS Symptoms Questionnaire with an open-ended question: "Did you experience anything unpleasant or painful during or after the study?". Three participants reported being more fatigued than usual after their TUS sessions. One of these three participants reported a mild headache the afternoon after their dACC TUS session, which resolved within a day, and no headache after the PCC TUS session. Another participant reported a persistent headache and neck pain after the sham session, which they attributed to having to remain still in the MRI rather than to the TUS procedure. They reported no symptoms after their TUS sessions. One participant reported a cool sensation ("as though my hair was damp") about an inch below where the transducer was placed during the PCC TUS session. This happened in the evening after the

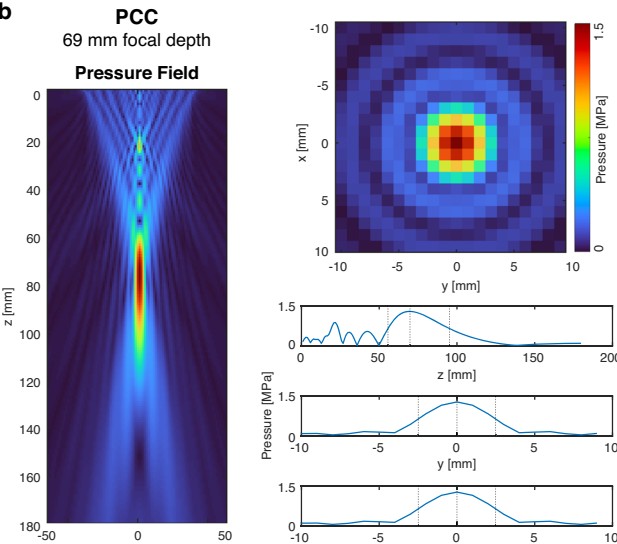

**Fig. 2 | Free-field acoustic simulations at $I_{SPPA} = 33.8$ W/cm$^2$.** The axial (z axis) and lateral (x and y axes) cross-sections of the acoustic pressure, as well as the pressure profile plots, are shown for two focal depths: **a** 60 mm, based on the average depth of the left dorsal anterior cingulate cortex (dACC) target, and **b** 69 mm, based on the left posterior cingulate cortex (PCC) target. In the pressure profile plots, the dotted lines represent the lower and upper bounds of the full width at half maximum (FWHM) of the ultrasound beam. At 60 mm, this corresponded to 32.1 mm along the axial plane of the beam, and 4.5 mm laterally. At 69 mm, the FWHM was 39.4 mm and 5.0 mm along the axial and lateral planes respectively. Source data are provided as a Source Data file.

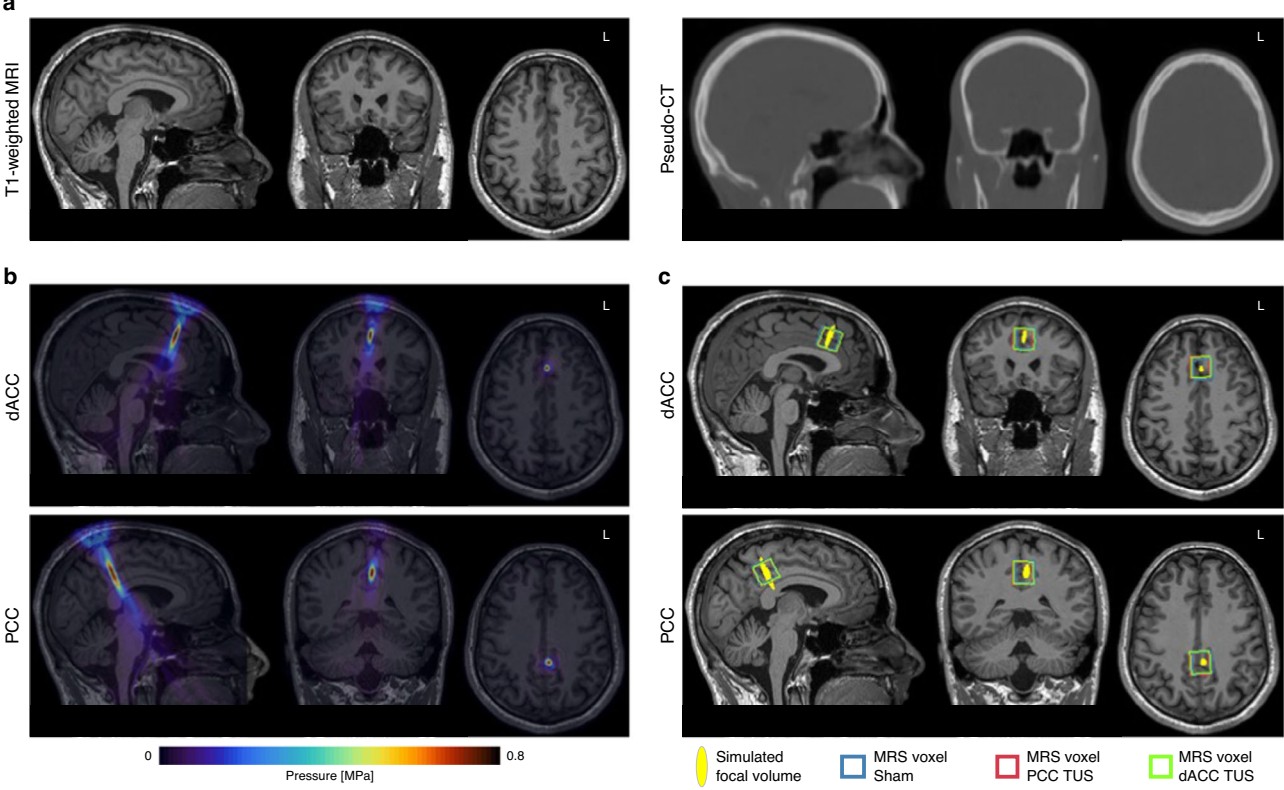

**Fig. 3 | Transcranial acoustic simulations in a representative individual. a** Pseudo-CT (right) derived from T1-weighted MRI (left) used to estimate skull acoustic properties. **b** Simulated ultrasound pressure field overlaid on the T1-weighted MRI showing reliable targeting of the dorsal anterior cingulate cortex (dACC) and posterior cingulate cortex (PCC). **c** Simulated transcranial ultrasound stimulation (TUS) focal pressure volumes shown with the $2 \times 2 \times 2$ cm³ MRS voxels for each session (sham in blue, PCC TUS in red and dACC TUS in green) in a representative individual. MPa megapascal.

TUS session and lasted for a few hours but was not described as unpleasant. The participant did not report any symptoms after their dACC TUS session. No other participants reported symptoms associated with TUS and were not able to distinguish between the TUS and sham sessions.

## TUS of the PCC selectively reduces GABA in the PCC
We found that TUS applied to the PCC region reduced GABA + /water (GABA + macromolecules, relative to water) in the PCC voxel, but not

**Table 1 | Simulated acoustic properties at the transcranial ultrasound stimulation (TUS) focus**

|  | dACC | PCC | t test p |
|---|---|---|---|
| Focal distance [mm] | 58.0 ± 4.9 | 69.5 ± 6.5 | 9.66 × 10⁻⁹ |
| Maximum pressure [MPa] | 0.66 ± 0.05 | 0.65 ± 0.04 | n.s. |
| MI | 0.95 ± 0.07 | 0.93 ± 0.06 | n.s. |
| $I_{SPPA}$ [W/cm²] | 15.07 ± 2.16 | 14.47 ± 1.78 | n.s. |
| $I_{SPTA}$ [mW/cm²] | 1507 ± 216 | 1447 ± 178 | n.s. |
| −6 dB focal volume [mm³] | 515 ± 197 | 745 ± 188 | 1.51 × 10⁻⁴ |
| Volume overlapping with MRS voxel [mm³] | 360 ± 75.8 | 421 ± 84.0 | 0.011 |
| Distance to COG of MRS voxel [mm] | 5.4 ± 3.0 | 7.1 ± 3.1 | n.s. |

Values are given as mean ± standard deviation. n.s. denotes non-significant two-sided *t* test. *dACC* dorsal anterior cingulate cortex, *PCC* posterior cingulate cortex, *MI* mechanical index, $I_{SPPA}$ spatial-peak pulse-average intensity, $I_{SPTA}$ spatial-peak temporal-average intensity, *MRS* magnetic resonance spectroscopy, *COG* centre-of-gravity.

in the dACC voxel, compared to sham (Fig. 4b). In the PCC voxel, the general linear model (GLM), with age, sex, simulated in situ $I_{SPPA}$ and TUS focal volume overlapping with the MRS voxel as covariates, showed a significant main effect of session, F(2,55) = 4.66, *p* = 0.013, $\eta^2$ = 0.141. Post-hoc comparisons were statistically significant for PCC TUS vs sham, t(55) = −2.88, *p* = 0.017, Cohen's *d* = −0.92, 95% confidence interval (CI) = [−1.578, −0.254] and PCC TUS vs dACC TUS, t(55) = −2.32, *p* = 0.048, Cohen's *d* = −0.73, 95% CI = [−1.383, −0.084], with no significant difference between dACC TUS and sham, t(55) = −0.57, *p* = 0.570, Cohen's *d* = −0.18, 95% CI = [−0.823, −0.458]. There were no significant differences between sessions in GABA + / water measured in the dACC voxel after dACC TUS (Fig. 4a), and no significant differences in Glx/water (the glutamate + glutamine complex, relative to water) between sessions in either voxel. These results show a localised decrease in GABA in the PCC after PCC TUS, suggesting a selective reduction in GABA only in the region that was sonicated.

## TUS increases functional connectivity with the targeted region
We first investigated changes in functional connectivity with the TUS target using a seed-based connectivity approach with a dilated mask of the TUS focal volume as the seed. Group-average maps of regions showing connectivity with the dACC and PCC seed during each run of the sham session are shown in Supplementary Fig. 1. These illustrate the relationship between the two regions and the classical salience network and default mode network for the dACC and the PCC respectively. We found no significant differences between the first and second rsfMRI runs during the sham session for either seed. Accordingly, whole-brain maps of connectivity with the TUS target were compared between

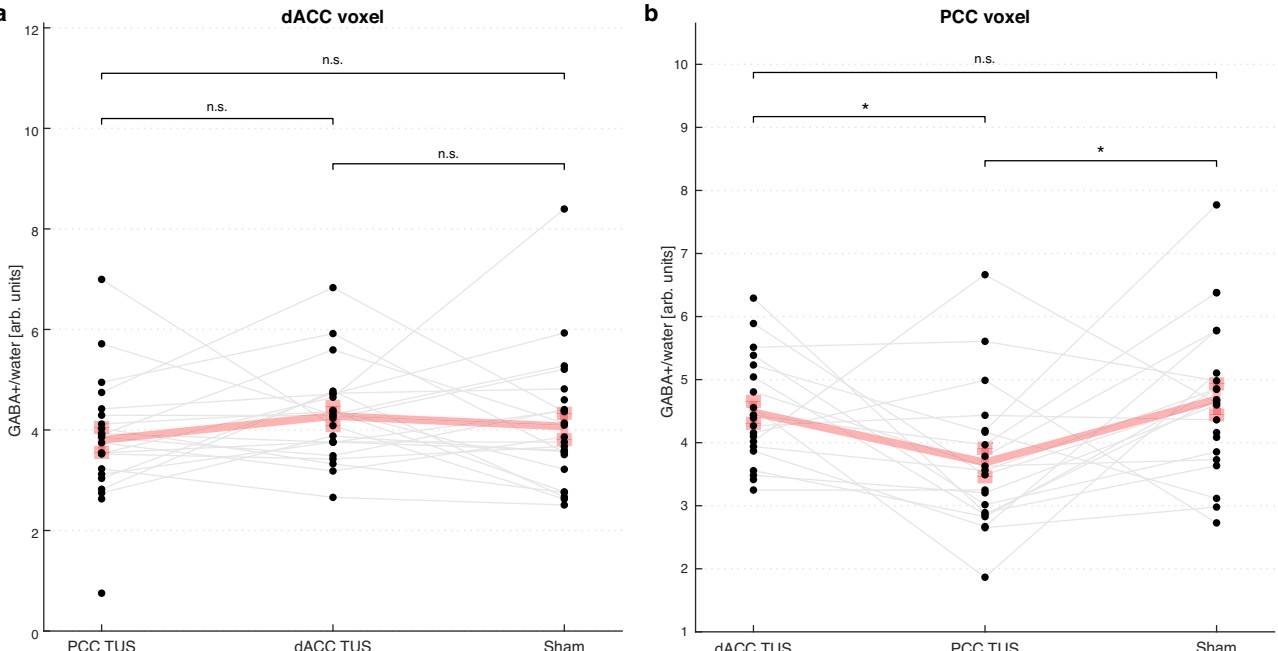

**Fig. 4 | Changes in GABA+/water after transcranial ultrasound stimulation (TUS).** Concentrations of GABA+/water in $n = 24$ individuals are shown in the **a** dorsal anterior cingulate cortex (dACC) voxel and **b** posterior cingulate cortex (PCC) voxel after each TUS session and sham. The active TUS session (i.e., when TUS was applied to and measured in the same region) is shown in the middle of both plots, to aid in the visual comparison against the sham and the control TUS sessions. The grey lines link measurements from the same individual across TUS sessions. The bold pink line represents the mean and standard error of the mean for each session. An ANCOVA and post hoc two-sided $t$ tests with Holm adjustment for

multiple comparisons were performed for each MRS voxel location (*$p < 0.05$, n.s.: not significant). ANCOVA in the dACC voxel did not show a significant main effect of session. ANCOVA in the PCC voxel showed a significant main effect of session, $F(2,55) = 4.66$, $p = 0.013$, $\eta^2 = 0.141$. Post-hoc comparisons were statistically significant for PCC TUS vs sham, $t(55) = -2.88$, $p = 0.017$, Cohen's $d = -0.92$, 95% Confidence Interval (CI) = [−1.578, −0.254], and PCC TUS vs dACC TUS, $t(55) = -2.32$, $p = 0.048$, Cohen's $d = -0.73$, 95% CI = [−1.383, −0.084], with no significant difference between dACC TUS and sham, $t(55) = -0.57$, $p = 0.570$, Cohen's $d = -0.18$, 95% CI = [−0.823, −0.458]. Source data are provided as a Source Data file.

each run of the TUS sessions and the average of the two runs during the sham session.

We found increases in functional connectivity with the dACC after both TUS of the dACC and TUS of the PCC compared with sham (all comparisons cluster corrected at $p < 0.05$). Functional connectivity of the dACC increased in the precuneus cortex approximately 13 mins after TUS of dACC compared with sham. Approximately 46 mins after TUS of the dACC, functional connectivity of the dACC was increased in a wider network of regions including the precuneus and intracalcarine cortex, bilateral thalamus, right putamen, left parahippocampal gyrus, and supplementary motor cortex including bilateral pre- and post-central gyri compared with sham (Fig. 5a).

Increases in functional connectivity of the dACC were also observed in the bilateral precentral gyri and the right superior parietal lobe 13 mins after TUS of the PCC compared with sham (Supplementary Fig. 2), which might indicate that PCC neuromodulation can affect brain connectivity beyond the region targeted. No significant differences in connectivity with the dACC seed were seen 46 mins after TUS of the PCC compared with sham.

Functional connectivity of the PCC increased only after TUS of the PCC compared with sham. In other words, the network profile of the PCC was not affected by TUS applied to another region, here the dACC. Functional connectivity was increased with the bilateral putamen at 13 mins, and with the precentral gyrus, bilateral auditory cortex and left temporal pole at 46 mins after TUS of the PCC compared with sham (Fig. 5b).

### TUS increases functional connectivity of the resting-state network associated with the target region

To investigate the effect of TUS on whole-brain networks at rest, we identified two well-defined resting-state functional connectivity

networks associated with our TUS targets using independent components analysis (ICA): the salience network, which has the dACC as a major component, and default mode network, of which the PCC is a major hub. Subject-specific maps of each network were obtained via dual regression and each run of each TUS session was compared against the average of the sham runs.

We found increased connectivity of the salience network after TUS of the dACC (Fig. 6a), and increased connectivity of the default mode network after TUS of the PCC compared with sham (Fig. 6b). Notably, these changes were only seen during the later run of rsfMRI (i.e., approximately 46 mins after TUS), consistent with the larger functional connectivity changes after TUS observed with the seed-based connectivity analysis. There were no significant changes in connectivity of the salience or default mode networks when TUS was applied to the other region.

### No associations between GABA and functional connectivity changes

We found no significant correlations between changes in GABA and functional connectivity changes, and no correlations between these changes and the focal volumes or intensity obtained from acoustic simulations.

### Discussion

Here we show neurochemical and functional connectivity changes after 80-seconds of offline theta-burst-patterned TUS in two deep cortical regions. We found consistent and robust changes in the PCC across MR modalities: decreased GABA as measured with MRS and increased rsfMRI functional connectivity with both the PCC target as well as with the default mode network. Our findings were less clear in the dACC, where we found only an increase in functional connectivity

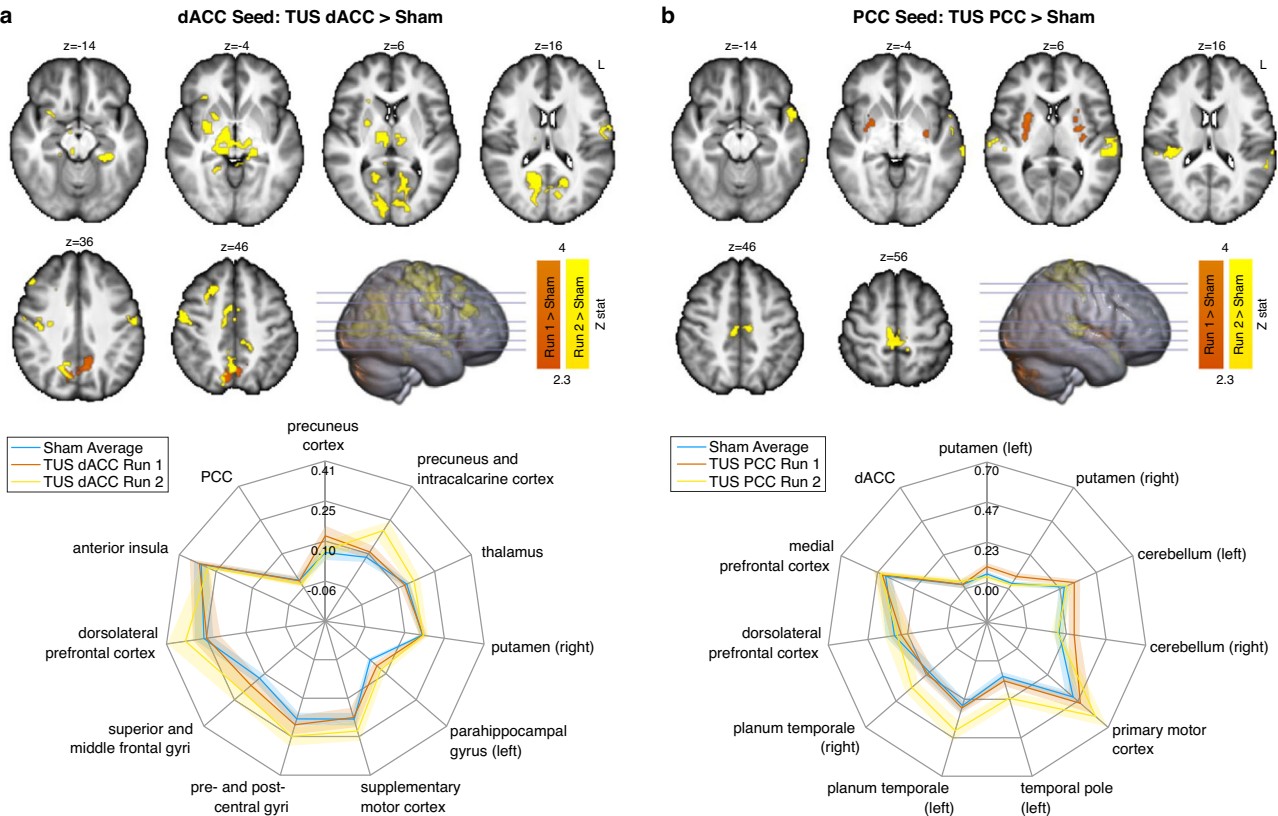

**Fig. 5 | Functional connectivity changes after transcranial ultrasound stimulation (TUS): seed-based connectivity analysis.** Whole-brain maps illustrate regions showing increased functional connectivity with **a** the dorsal anterior cingulate cortex (dACC) seed after TUS was applied to the dACC, and **b** the posterior cingulate cortex (PCC) seed after TUS was applied to the PCC, compared with sham. For each seed region, a whole-brain mass-univariate GLM was performed and the $Z$ statistic images (thresholded using clusters determined by $Z > 2.3$ and a FWER corrected cluster significance threshold of $p = 0.05$) from the two one-sided contrasts (i.e., fMRI Run 1/2 vs. the average of the sham runs) are shown for statistically significant clusters. Clusters in orange represent regions with significantly higher functional connectivity at ~13 mins after TUS (i.e., fMRI Run 1), and clusters in yellow show regions with significantly higher functional connectivity at ~46 mins after TUS (fMRI Run 2) compared with the average of both sham runs. For each seed region, the spider plots show the functional connectivity (parameter estimate from the GLM) during each run sampled from the regions showing significantly increased connectivity and three control regions: (i) the region not targeted with TUS (i.e., for the dACC seed, this would be the PCC, and vice versa), (ii) a region known to be highly connected to the seed region (the anterior insula for the dACC seed, and the medial prefrontal cortex for the PCC seed), and (iii) the dorsolateral prefrontal cortex. The error bars show the standard error of the mean. Whole-brain maps are overlaid on the average T1-weighted MRI of all participants. Source data are provided as a Source Data file.

with the target and with the salience network, but no changes in GABA. Taken together, these changes suggest that theta-burst TUS can transiently decrease cortical inhibition in deep cortical regions in humans for at least 50 mins after TUS. Our findings complement existing evidence that theta-burst TUS increases corticospinal excitability in the human motor cortex[18], and additionally represent evidence of neurochemical and functional connectivity changes associated with theta-burst TUS in deep cortical regions. The timescale of excitability changes, up to at least 50 mins after TUS, relative to the duration of stimulation applied suggests induction of reversible neuroplasticity, possibly linked to long-term potentiation/depression of neurons.

The significant decrease in GABA levels in the PCC voxel after theta-burst TUS was applied to the PCC compared to both sham and TUS applied to the dACC, suggests a localised decrease in GABA within the targeted region. This suggests a TUS-mediated reduction in GABA in humans, and complements findings of extracellular GABA decreases in rats following ultrasound[14]. Localised decreases in GABA have been reported after other types of stimulation, including TMS and tDCS. In humans, studies using repetitive TMS with MRS (for a review, see[12]) have shown changes in GABA levels at the TMS target[20,21] and also in network regions connected to the target[22,23]. In a study using anodal tDCS, GABA was found to gradually decrease during the 20-minute stimulation duration, with the largest decrease found at around 10-

15 mins post-stimulation before gradually returning to baseline[24]. Here, we show that the 80-second theta-burst TUS protocol induces GABA decreases that persist up to at least 30 mins post-stimulation, however, because we only sampled one voxel at the TUS target location during each session, it is unknown how GABA levels change during and immediately after stimulation, and how long it would take for GABA levels to return to baseline after TUS.

Functional connectivity of the PCC was increased in a network of whole brain regions after TUS of the PCC, complementing the decreased GABA (or decreased inhibition) found with MRS. Functional connectivity of the dACC was also increased after TUS of the dACC, although no corresponding changes in GABA were found. In both the seed-based and network-based analyses, there were differences in the pattern of increased connectivity between the early and late rsfMRI runs (approximately 13- and 46-mins post-TUS respectively), with a larger network of regions showing increased connectivity in the late rsfMRI run. This was reflected in the network-based ICA where both the default mode network and salience network showed significant changes from sham after TUS of their associated region, only in the late rsfMRI run and not the early run.

Using TMS, magnetoencephalography (MEG) and the same theta-burst TUS protocol, Samuel and colleagues[25] reported increased motor cortex excitability as well as increased local MEG coherence in motor regions, with TMS changes being correlated with the MEG coherence

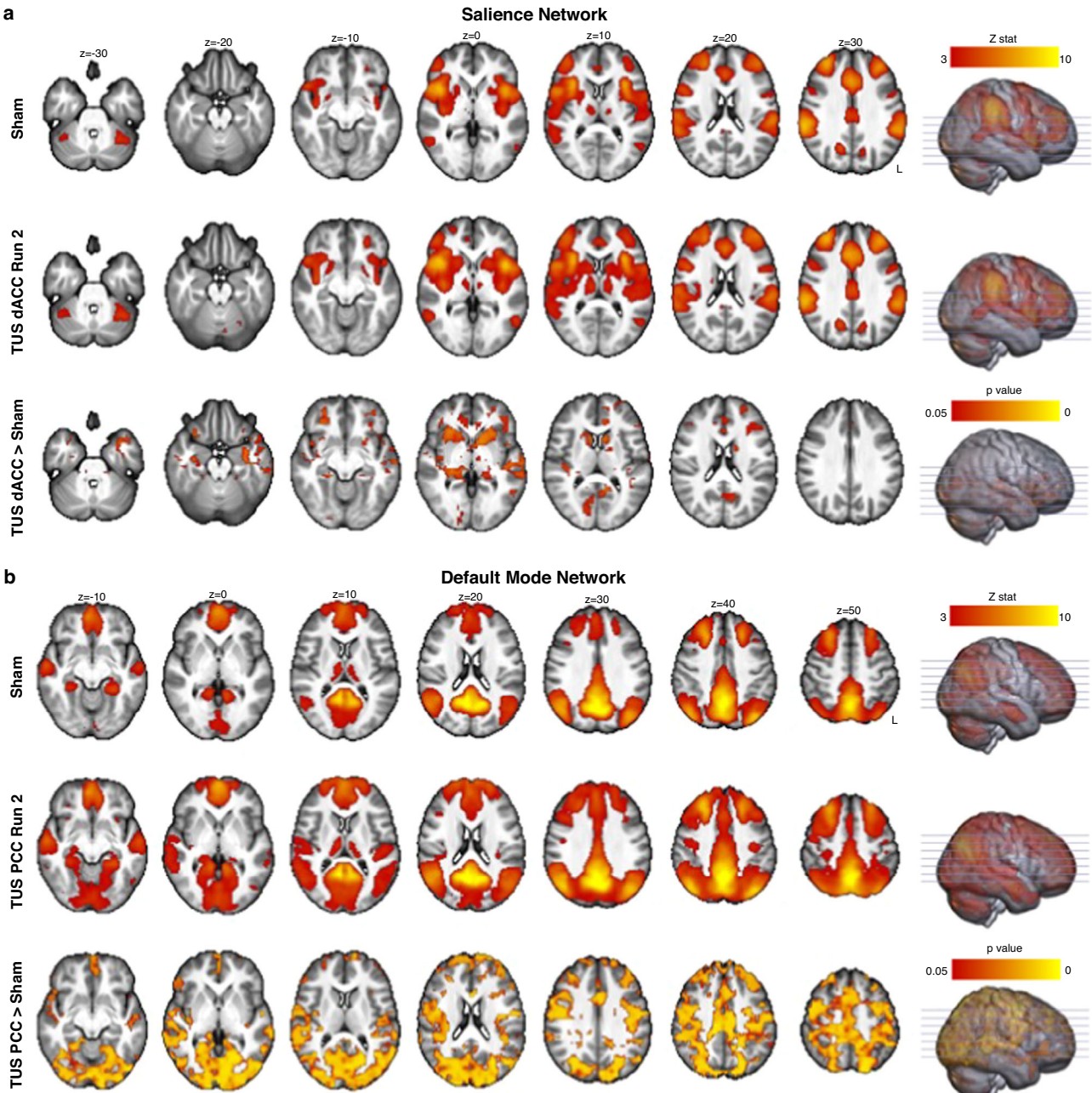

**Fig. 6 | Functional connectivity changes after transcranial ultrasound stimulation (TUS): independent components analysis.** Group-average maps of the **a** salience network and **b** default mode network identified using independent components analysis on fMRI runs during sham sessions are shown in the top row of each sub-panel. The middle row of each sub-panel shows the group-average network during the fMRI run ~46 mins after TUS (i.e., fMRI run 2) obtained via dual regression. The bottom row shows the spatial map of significant differences in connectivity of the networks between run 2 of the TUS sessions and the average of sham sessions. dACC dorsal anterior cingulate cortex, PCC posterior cingulate cortex.

measures. Several studies have reported fMRI changes with "online" TUS predominantly in the regions targeted with TUS[26,27], however relatively few studies have reported fMRI or blood flow changes with offline TUS protocols thought to induce longer-term changes in cortical excitability. In one such study, TUS applied to the right inferior frontal gyrus decreased functional connectivity in a network of regions related to emotion and mood regulation[9]. Another targeting the globus pallidus with an inhibitory TUS protocol found decreased connectivity in a network of frontoparietal and thalamic regions[10]. Our results complement these studies and show distal changes in a network of brain regions functionally related to the TUS target. We did not find an association between functional connectivity and GABA

changes, which could be due to different mechanisms underlying GABA-mediated localised decreases in cortical inhibition and increases in functional connectivity in the network of regions distal but functionally connected to the targeted region.

Our results also suggest a possible state-dependent mechanism underlying TUS neuromodulation. We saw a pattern of more robust effects of neuromodulation when TUS was applied to the PCC, a region known to be important during wakeful rest. Local changes in GABA were only found with TUS applied to the PCC but not with TUS of the dACC, and although TUS of both regions showed functional connectivity changes in their respective networks, TUS of the PCC additionally increased functional connectivity of the dACC seed. The PCC is

a major component of the default mode network, which is known to be more "active" during rest, while the dACC is part of the salience network, which is typically found to be anti-correlated with the default mode network. There is growing evidence for a state-dependent mechanism in brain stimulation[28], and it is increasingly accepted that the cognitive state or state of consciousness has an important influence over how the brain will respond to interventions. In patch-clamp recordings in CA1 pyramidal neurons of rodent hippocampal brain slices, ultrasound has been shown to either inhibit or potentiate neuronal firing depending on the firing regime (e.g. high frequency, irregular/infrequent or absent firing) of the cells targeted[29]. Similarly, in macaques, the modulatory effects of TUS differ depending on whether the neurons are active or at rest[30].

The inherent differences in cortical morphology and composition of neurons in the PCC and dACC could also contribute to the difference of neuromodulatory effects found. The dACC is a complex region and 70% of individuals show an additional cingulate sulcus, the paracingulate sulcus, in addition to the cingulate sulcus in at least one hemisphere[31]. This could contribute to more heterogeneous function of the dACC and similarly a heterogeneous response to neurostimulation. There is also the possibility that stimulation exhibits distinct neuromodulatory effects in different neuron populations[32]. Transcriptomics data from the Human Protein Atlas[33] suggest potential tissue-composition differences between the two regions, specifically variation in the presence of several ion channels. T-type $Ca^{2+}$ channels for example are thought to be sensitive to sonication[3] and corresponding protein-coding genes may be expressed preferentially in the PCC compared to the dACC[33].

We found no changes in the concentration of Glx (the glutamine + glutamate complex). There could be several explanations for this. Firstly, MEGA-PRESS is a GABA editing sequence and not optimised for measuring Glx. Glx may be quantified from off-resonance MEGA-PRESS spectra, but it is unclear how reliable these measurements are in different brain regions. One study has found that PRESS and off-resonance MEGA-PRESS Glx estimates are highly correlated in the dorsolateral frontal cortex[34]. However, another study specifically looking at the dACC found that there was better agreement between PRESS and off-resonance MEGA-PRESS in the sensorimotor cortex than in dACC, although both regions showed poor agreement with separately acquired PRESS spectra. Future studies could use other sequences to quantify Glx or acquire spectra at higher MR field strength so that glutamine and glutamate signals can be measured separately.

Our acoustic simulations show differences in the pressure profile and size of the focal field at different target depths, which is an important consideration when targeting deep cortical regions and highlights the importance of acoustic simulations in TUS. We saw interindividual variability in terms of focal volume (~200 mm³, or 27–37% of the average focal volume for PCC and dACC, respectively) and intensity (approximately 3 W/cm², or 12–13% of the average intensity) at the TUS targets. This inter-individual variability could be due to several factors including scattering of the acoustic pressure due to the skull and positioning of the TUS relative to the individual's skull. Although we did not find any associations between the variability of focal volume or intensity with the amount of change in GABA or functional connectivity, this may be different in more difficult-to-target regions (e.g., where the skull is not strictly perpendicular to the trajectory of the beam, or is heterogeneous in its composition), and is worth considering or accounting for when analysing the results.

Another possible limitation is that the study was designed as a single-blind study: the experimenters and the researchers who analysed the data were aware of the study conditions, which could possibly introduce some bias during the data collection and analysis. We tried to design the sound delivered via bone-conduction headphones during the sham condition to closely match the TUS

condition. A small number of participants ($n = 4$) who were familiar with the TUS equipment were able to identify the sham condition. However, the majority participants ($n = 20$) were unaware that we had a sham condition when asked after completing the study. Thus, we believe that the majority of participants were sufficiently blinded to the study conditions.

At present, there is interest in the TUS research community to identify TUS protocols that are either excitatory or inhibitory. Our findings help shed light on this process by providing an explanation for how theta-burst TUS induces neuroplasticity in two deep cortical targets and their associated networks of whole-brain regions and suggest that these changes may be state-dependent. This has fundamental implications for the understanding and design of both basic TUS research and its clinical translation.

## Methods

The study was preregistered on the Open Science Framework on 10th March 2022: https://osf.io/bcf4v. The only deviation from the preregistered protocol is that we set the free-field $I_{SPPA}$ based on our hydrophone measurements. Data supporting the findings of this study are available at https://osf.io/rp5g4/.

### Participants

Twenty-four healthy volunteers (14 female) aged between 22 and 53 years (mean = 33.8, s.d. ± 9.7) participated in the study. Participants reported no current diagnosis of neurological or psychiatric disorders and were not taking any medications known to affect brain excitability at the time of the study. Specifically for TUS and MRI safety, we excluded participants who at the time of the study: 1) were pregnant (self-reported), 2) were using psychoactive drugs, 3) had any contraindication to MRI, 4) had a current or previous diagnosis of any neurological disorders, 5) had a current or previous diagnosis of psychiatric disorders (including enduring severe mental illness but excluding history of depression/anxiety), 6) had a first-degree relative with epilepsy, 7) experience extreme mood fluctuations, or 8) were currently using prescription or non-prescription medication (including drugs acting on the central nervous system, pro-convulsive drugs, drugs (or combinations of drugs) that lower seizure threshold or withdrawal from drugs that incurs a lowering of seizure threshold (i.e. anti-convulsant withdrawal); for a full list of excluded medication, see Supplementary Table 3), unless these did not interfere with study procedures or compromise safety. The study was approved by the University of Plymouth Faculty of Health Staff Research Ethics and Integrity Committee (reference ID: 2487; date: 13/12/2021). Written informed consent was obtained from all participants after experimental procedures were explained in full. Participants were compensated £30 for completing each session and travel expenses up to £10 per session. All study sessions took place at the Brain Research and Imaging Centre in Plymouth, United Kingdom.

### Study design

Figure 1 summarises the study design and procedures. All participants completed three separate TUS and MRI sessions at least one week apart and at the same time of the day for each participant (± 30 mins) to control for the effects of the circadian rhythm on GABA fluctuations. During each session, they underwent TUS applied to either the left dACC, the left PCC, or sham TUS, followed by a series of MRI scans. During the sham TUS session, no stimulation was delivered, and the transducer was positioned over the mid-cingulate cortex. The order of the three sessions (dACC TUS, PCC TUS, or sham TUS) was counterbalanced across subjects.

For participants who were assigned to have verum TUS as their first session, we acquired a high-resolution T1-weighted MR image prior to their first TUS and MRI session. The high-resolution T1-weighted MR image was used to estimate each participant's skull

model of bone density and geometry for use in acoustic simulations and for neuronavigation.

## Ultrasound stimulation

We used the NeuroFUS TPO and CTX-500-4 transducer (Brainbox Ltd., Cardiff, UK). This consisted of a four-element ultrasound transducer (64 mm diameter) with a central frequency of 500 kHz. We used the theta-burst TUS protocol[18] with the following parameters: pulse duration = 20 ms, pulse repetition interval = 200 ms and total duration = 80 s, giving a total of 400 pulses. The target free field spatial-peak pulse-average intensity ($I_{SPPA}$) was kept constant at 33.8 W/cm² for each participant. We performed transcranial acoustic simulations (see "Acoustic simulations" section) to ensure that we remained below the FDA guidelines for diagnostic ultrasound (MI ≤ 1.9; $I_{SPPA}$ ≤ 190 W/cm²) after transcranial transmission. In addition, we ensured that the maximum temperature rise across the entire 80 s duration of TUS did not exceed 2 °C in all our thermal simulations.

We prepared each participant's head by parting any hair over the intended target and applying ultrasound transmission gel (Aquasonic 100, Parker Laboratories Inc.). We applied ultrasound gel to the transducer, used a gel pad (Aquaflex, Parker Laboratories Inc.) and, as far as practically possible, ensured no air bubbles between the transducer face and participant's head.

Neuronavigation was performed with the Brainsight software v 2.4.11 (Rogue Research Inc., Montréal, Québec, Canada) on anatomical T1-weighted MRI scans from each participant. The focal depth was adjusted for each participant and brain region based on the neuronavigated target. During TUS, we sampled the transducer coordinates with the software and noted any deviations from the intended focus. The positions of the transducer and target were used in acoustic simulations. After each TUS sonication, participants were asked to report any symptoms they think were associated with TUS via an open-ended TUS symptoms questionnaire completed the following day.

Sham TUS was delivered in the same way as verum TUS, except that the power to the transducer was turned off. To control for auditory effects, we played a sound mimicking the pulse repetition and duration of verum TUS via bone-conduction headphones. We designed the waveform to match the ultrasound pulses produced by the TUS protocol (frequency, pulse repetition frequency and pulse duration). We adjusted the sampling rate to change the pitch of the sound and asked five lab members who have experienced the TUS protocol to vote on the sound that best matched the TUS and played this during the sham condition. Headphones were placed on the participant's head ~2 cm posterior to the temples for all the sessions, and the sound was only played during the sham session so that participants do not hear two separate sounds (i.e., the actual sound from the TUS protocol and the audio played through the headphones) during verum TUS conditions. When they had completed all three sessions, participants were asked if they could distinguish between sham and verum TUS.

## Magnetic resonance acquisition

Immediately following TUS, participants underwent a series of MRI scans on a Siemens MAGNETOM Prisma 3 T scanner (VE11E, Siemens Healthineers, Erlangen, Germany) with a 32-channel head coil. The sequence of scans was as follows:

1. T1-weighted magnetisation-prepared rapid gradient echo (MPRAGE) sequence acquired in the sagittal plane for MRS voxel planning (2100 ms repetition time (TR), 2.26 ms echo time (TE), 900 ms inversion time, 8° flip angle (FA), GRAPPA acceleration factor of 2, 256 × 256 mm field of view, 176 slices and 1 mm³ isotropic voxels)
2. Localiser (to check for movement relative to T1-weighted MR scan
3. 5-minute resting-state gradient echo echo planar imaging (GE-EPI) fMRI scan during which the MRS voxels were positioned (Centre

for Magnetic Resonance Research, CMRR, multiband fMRI sequence from https://www.cmrr.umn.edu/multiband/, acquisition plane approximately parallel to the AC-PC line, 2000 ms TR, 30 ms TE, 74° FA, 2.5 mm slice thickness, no slice gap, multiband acceleration factor of 2, and 60 interleaved slices of 80 × 80 matrix size, giving a voxel size of 2.5 × 2.5 × 2.5 mm³)
4. pre-MRS localiser
5. MRS flip angle calibration (with voxel placed on TUS target region)
6. MRS acquisition in TUS target region (2 × 2 × 2 cm³ voxel, CMRR single-voxel spectroscopy MEGA-PRESS sequence[35], 2000 ms TR, 68 ms TE, with VAPOUR water suppression, 128 averages, Edit On/Off frequency at 1.90/7.50, editing pulse bandwidth 50.55 Hz, number of spectral points 2048, spectral width 1850 Hz, water unsuppressed reference: 16 averages)
7. MRS acquisition in control region (with same parameters as above)
8. post-MRS localiser
9. field map (all measurement parameters made to match fMRI acquisition as far as possible).
10. 5-minute resting-state fMRI (with same parameters as above)

Automatic shimming (Siemens Brain method) was performed before each MRS acquisition, with additional manual shimming applied if the full-width half-maximum of the signal was over 20 Hz.

## Target location for ultrasound and MRS

The left dACC and left PCC targets for TUS were identified based on an initial co-registration with the Montreal Neurological Institute (MNI) coordinate space at $x = -5$, $y = 24$, $z = 30$ for the dACC and $x = -5$, $y = -35$, $z = 35$ for the PCC. This was then adjusted based on anatomical landmarks on each individual's T1-weighted MRI. The dACC target was aligned with the back of the genu and superior-most point of the body of the corpus callosum, centred on a patch of grey matter in the cingulate gyrus. The PCC target was aligned with the middle of the splenium of the corpus callosum, roughly in line with the ascending ramus of the cingulate sulcus and centred on a patch of grey matter just anterior to this, below the cingulate sulcus. MRS was acquired in a voxel centred on the target from the TUS session to ensure overlap between the TUS focus and MRS acquisition. Examples of voxel placement in two individuals are shown in Fig. 3c and Supplementary Figure 3.

## Acoustic measurements and simulations

We measured the output of our NeuroFUS transducer in a custom-built tank[36] (https://github.com/SamC873/FUSF_Hydrophone_Scanner) using a calibrated 0.4 mm Onda hydrophone (HGL-0400 S/N: 2498). The tank (a glass aquarium; length = 80 cm, width = 35 cm) was filled with deionised water (water depth = 27 cm) at 23.1 °C. The NeuroFUS system was set to produce a waveform with 0.06 ms pulse duration, 2 ms pulse repetition interval at a maximum of 2 Watts power output. Line scans were performed with 1 mm steps along the beam axis over a range of 80 mm centred on the focus, and with 0.5 mm steps over a range of 30 mm along the lateral cross-sections of the ultrasound beam passing through the focus. We measured the axial and lateral pressure profiles for two focal depth settings (60 mm and 69 mm) and based our acoustic simulations on the measured free-field intensity.

We used the k-Wave Toolbox[37] (version 1.4) in MATLAB (R2020b, MathWorks, Inc.) for our simulations and modelled our transducer based on the physical properties of the NeuroFUS transducer and phases reported on the TPO unit. We first performed acoustic simulations in water to characterise the ultrasound beam before transcranial attenuation for a target $I_{SPPA}$ of 33.8 W/cm². Since the dACC and PCC are at different depths in the cortex, we simulated the ultrasound beam for a focal depth of 60 mm and 69 mm,

representing the average focal depths of the dACC and PCC targets across all individuals.

Next, we performed transcranial simulations for the dACC and PCC for each participant in the study. We estimated the skull for each participant from a pseudo-CT derived from the participant's T1-weighted MRI using a deep learning method[38,39]. The skull was obtained from pseudo-CT images by thresholding at 300 HU and clamping values above 2000 HU. Pseudo-CT HU intensities were linearly mapped to acoustic properties using equations for density, speed of sound and absorption coefficient as described in refs. 40,41. We set our simulation grid size to a $256 \times 256 \times 256$ matrix centred on the midpoint between the transducer and focus with a grid spacing of 0.5 mm (i.e. 6 points per wavelength at 500 kHz). Our acoustic simulation methods are described in further detail elsewhere[38] and the code is available online[42].

## Spectroscopy data analysis
MRS processing and analysis were performed in Gannet[43] (http://www.gabamrs.com/). Processing steps included 3 Hz line broadening, correction for frequency and phase errors by spectral registration[44], outlier rejection, time averaging, and eddy current correction. The edited difference spectrum was modelled to quantify the 3.0 ppm GABA+ and 3.75 ppm Glx signals relative to water. The T1-weighted MR image was segmented using SPM12[45] to obtain tissue-corrected measurements[46] within the MRS voxel. MRS spectra were visually inspected for spectral artifacts, including lipid contamination, subtraction errors and a non-constant baseline. We excluded data if they were outliers on the following quality metrics: FWHM, GABA+ signal-to-noise ratio (SNR), linewidth, and model fit errors. Example spectra acquired from the dACC and PCC voxel during a Sham session are shown in Supplementary Fig. 3.

For each voxel, changes in GABA + /water and Glx/water between both TUS and sham sessions were assessed with a GLM with age, sex, simulated in situ $I_{SPPA}$, and simulated TUS focal volume within the MRS voxel as covariates (ANOVA with main effects and post hoc tests, using the Holm correction for multiple comparisons, considered statistically significant at $p < 0.05$). Covariates were chosen based on factors that are likely to affect GABA levels within the voxel. Statistical analyses were performed in JAMOVI version 2.0.0 (https://www.jamovi.org). Missing or excluded data are treated as NaNs in the model.

## Functional MR data analysis: seed-based connectivity
FMRI data were pre-processed and analysed using FEAT (FMRI Expert Analysis Tool) Version 6.00, part of FSL (FMRIB's Software Library, www.fmrib.ox.ac.uk/fsl). Pre-processing included motion correction, B0 field inhomogeneity correction, brain extraction, spatial smoothing (5 mm FWHM) and highpass filtering (0.01 Hz). rsfMRI data were co-registered to the MNI standard space via a linear transform to the subject's high-resolution T1-weighted MRI and a non-linear transform to the MNI template. Motion outliers were identified using the fslmotionoutliers tool and were included as nuisance covariates along with the average signal from the white matter and cerebrospinal fluid, and the six motion parameters from the motion correction step.

For each subject and each session, a seed-based connectivity analysis was performed with the subject-specific TUS focal volumes obtained from the dACC and PCC acoustic simulations as seeds. To create the subject-specific TUS seed, a binary mask was first created from the top 25% maximum pressure intensities in the simulated pressure field. This binary volume was then dilated by two voxels to give an average seed volume of $805 \pm 162$ mm³ for the dACC and $1003 \pm 178$ mm³ for the PCC (for reference, a typical 6 mm radius spherical seed used in seed-based functional connectivity analyses had a volume of 905 mm³). The average timeseries was sampled from each

seed and used as the variable of interest in a voxel-wise whole-brain GLM implemented using FSL's FEAT, with the nuisance regressors described above as variables of non-interest.

Functional connectivity of the dACC and PCC seed were first combined at the subject level, comparing each TUS session and run against the mean of the sham runs with a fixed effects model. Comparisons across subjects for each seed were done using a mixed-effects model (FLAME1 + 2) with automatic outlier detection with age and sex as covariates. Whole-brain Z statistic maps were thresholded using clusters determined by $Z > 2.3$ ($p = 0.05$) and a familywise error-corrected cluster significance threshold of $p = 0.05$. Comparisons of each TUS session against the corresponding run in the sham condition (i.e. run 1 TUS vs run 1 sham and run 2 TUS vs run 2 sham) yielded results which were similar in spatial extent but not statistically significant (Supplementary Figure 4). Since there were no statistically significant differences between the two runs in the sham condition, we combined the sham runs to increase the fMRI signal-to noise ratio and our power to detect differences between active TUS and sham conditions.

## Functional MR data analysis: resting-state network connectivity
We investigated the effect of TUS on two brain networks of interest at rest involving our two target brain regions. These were 1) the salience network, comprising the dACC and anterior insula, and 2) the default mode network, comprising the PCC, medial prefrontal cortex and bilateral angular gyri.

We first identified the group-average spatial maps of the networks of interest using independent components analysis (multi-session temporal concatenation in FSL MELODIC; https://fsl.fmrib.ox.ac.uk/fsl/fslwiki/MELODIC) of the sham sessions only. We then used a dual regression approach[47] to generate subject-specific versions of the group-average spatial maps and associated timeseries for each subject, session and run. Briefly, this involved regressing the group-average set of spatial maps (as spatial regressors in a multiple regression) onto each subject's 4D space-time dataset, giving a set of subject-specific timeseries, one per group-level spatial map. Next, those timeseries are regressed (as temporal regressors in a multiple regression) into the same 4D dataset, resulting in a set of subject-specific spatial maps, one per group-level spatial map. We then tested for session differences using FSL's randomise permutation-testing tool and 5000 permutations.

## Exploratory analyses: relationship between GABA and functional connectivity changes and associations with simulated in situ TUS intensity
We investigated whether the TUS-mediated changes in GABA and functional connectivity of the PCC were correlated using Pearson's correlations. First, we sampled the mean functional connectivity strength within regions showing a significant difference in connectivity for each individual rsfMRI run. The difference in functional connectivity between PCC and sham runs was then correlated against the difference in GABA between PCC and sham sessions. Correlations were assessed for significance at the conventional alpha value of $p < 0.05$.

The skull accounts for a large amount of attenuation and aberration of the TUS intensity at the target location and the amount of attenuation varies between individuals based on skull structure and depth of target. We explored the association between simulated TUS intensity and focal volume and MRS and rsfMRI measures across individuals using Pearson's correlations as above.

## Reporting summary
Further information on research design is available in the Nature Portfolio Reporting Summary linked to this article.

## Data availability

The raw and processed MR data and acoustic simulation data generated in this study have been deposited in the Open Science Framework database under the CC-By Attribution 4.0 License: https://osf.io/rp5g4/. Human Protein Atlas data was accessed via the online portal: https://www.proteinatlas.org/humanproteome/brain/cerebral+cortex. Source data are provided with this paper.

## Code availability

The code for generating pseudo-CT from T1-weighted MR images and for running the acoustic simulations as described in this work are available on GitHub: https://github.com/sitiny/mr-to-pct and https://github.com/sitiny/BRIC_TUS_Simulation_Tools.

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

## Acknowledgements

The authors thank Drs. Nadège Bault and Matt Roser for their help with setting up scanning sequences, Ms. Sarah Shahrin for assistance with uploading data supporting this work to the Open Science Framework, all study participants for taking part in the study, and members of the Fouragnan lab for helpful discussions. This research was supported by a UKRI Medical Research Council Future Leaders Fellowship grant MR/T023007/1 (to E.F.F.). Scanning for this study was supported by the Brain Research & Imaging Centre (BRIC) Proof of Concept Support funding (to S.N.Y. and E.F.F.).

## Author contributions

S.N.Y and E.F.F. conceived this research and designed the study. L.V., C.S., E.M., and S.H. contributed to the study design and advised on data quality and analysis. J.R. and S.N.Y. planned and tested the imaging sequences. J.R. optimised the sequences and advised on MRS physics and data quality. E.M. advised on ultrasound physics, transducer measurements, and acoustic simulations. S.N.Y., T.A.W., and E.F.F. conducted the study and analysed the data. S.N.Y. and E.F.F. wrote the manuscript with input from all authors. All authors reviewed the final manuscript.

## Competing interests

The authors declare no competing interests.
