## [Peer Review File · Nature Communications]

Transcranial focused ultrasound-mediated neurochemical and functional connectivity changes in deep cortical regions in humansReviewer #1 (Remarks to the Author):

This study used a theta burst transcranial ultrasound stimulation (TUS) protocol to stimulate the posterior cingulate (PCC) and dorsal anterior cingulate cortex (dACC) in 24 healthy subjects. It was found that functional connectivity measured by fMRI increased in both stimulated regions but mostly at a later time point of ~46 min rather than an earlier time point of ~13 min after TUS. GABA levels measured by MR spectroscopy was reduced in the PCC and not the dACC after TUS. The authors concluded that the effects of TUS are state dependent.

This is an interesting and novel study on the effect of theta burst TUS. There are limitations to the study such as the size of the ultrasound field and the overlap with MRS voxel was different for the two targets. There are other issues that need to be considered.

Major comments

1. The conclusion that the effect of TUS is state dependent is not convincing. The main argument is that the PCC is active during wakeful rest. However, another (perhaps more likely) possibility is that the effects are location dependent. Possibilities other than state dependency was not mentioned in the Abstract or the Introduction.
2. Sound was played to the subjects only in the sham condition. How the sound was set to be similar to the auditory effects of TUS was not clear. This makes the conditions different for the real versus sham TUS and could potentially account for some of the results. It is not clear why the sound was not played in the real TUS conditions.
3. The subjects were obviously not blinded to the study conditions. I assume that the experimenters and the investigators who analyzed the data was not blinded. This should be stated and acknowledged as a limitation of the study.
4. In the fMRI analyses, the first and second runs were compared to the average of the two runs in the sham session. However, there could be non-specific effects of time in the study. The first run in the real TUS conditions should be compared to the first run of the sham condition, and the second run in the real TUS conditions compared to the second run of the sham condition.

Minor comments

5. Based on the Figure 3 and the coordinates provided, the left PCC and dACC were targeted but this was not stated in the manuscript.
6. Pg. 18 What "the regime of the cells" mean is not clear.
7. Pg. 20. It is not clear what categories of mediations were not allowed.
8. A previous study (PMID: 36228977) has shown that there are wide spread changes in brain oscillations and connectivity with theta burst TUS to the motor cortex. This should be discussed in relation to the current findings.

Reviewer #2 (Remarks to the Author):

The authors used the new technique of transcranial ultrasound sonification (TUS) to stimulate deep cortical areas dACC and PCC and compared this stimulation with a sham stimulation. dACC is the crucial area of the salience network and PC is the crucial area of the default mode network. The stimulation protocol consisted of a theta-burst stimulation for 80 seconds. The simulation of the stimulation effects confirmed focal stimulation with an increase of temperature at the stimulation sites below 2° C. The measurement of GABA concentration at the sites of stimulation revealed that the GABA concentration decreased in PCC, but not in dACC. The stimulation of dACC increased the activity of the salience network and the stimulation of PCC increased the activity of the default mode network. The ROI based connectivity analysis resulted also in increased connectivity of both areas. The effects of the TUS on neuroplasticity lasted for at least 50 minutes and was apparently state dependent.

The study includes the novel technique of TUS and sheds new light on the effects of this non-invasive stimulation technique. The study design is elaborate and sound. The involved brain imaging techniques are high standard. The methods of data analysis are elaborated and the result are plausible and very interesting.

I have only two questions:

1. A question concerning the theta-burst stimulation protocol. The stimulation consisted of 20 ms pulses, spaced by 200 ms pause. It seems to me that the authors used a continuous theta-burst protocol. It is known from the transcranial magnetic stimulation that continuous theta-burst stimulation is inhibitory and intermittent theta-burst stimulation is excitatory. How can the authors be sure that the protocol they used was excitatory?

2. The GABA concentration decreased significantly in PCC, but not in dACC. For the assessment of the proton magnetic resonance spectroscopy (MRS) ROIs of 2 x 2 cm, which is rather low for a 3T MRS. Would the significance increase if the authors used for example 3 x 3 cm voxels?

Response to Reviewers

We thank the Reviewers and the Editor for their helpful and constructive comments. We have carefully considered all the issues raised and made extensive changes to the manuscript. We hope that they will all agree with our feeling that the manuscript has been considerably strengthened as a result.

We have responded to each of the points made individually below, including any relevant excerpts from the manuscript in each case. All changes to the manuscript and supplementary information have been marked in red in the revised documents.

In addition to the reviewers' comments, we have adjusted our TUS acoustic simulations to reflect the output of our transducer as measured with a hydrophone in a water tank (main changes highlighted on pg. 7, 25-26, and Table 1). The rationale behind this is that the equipment we used has not yet been CE marked so we wanted to obtain a measurement to ensure reproducibility and safety, in line with recommendations set out by experts in acoustic metrology. This was achieved in collaboration with Dr Eleanor Martin who leads the ITRUSST equipment working group (<https://itrusst.github.io/documentation/equipment.html>). Based on our hydrophone measurements, we have amended the simulated intensities reported in our manuscript. This did not change any of our main results. We hope that you will allow us to include these changes as we believe that they serve to strengthen the paper and provide a solid basis for the results we present. We would thus like to include Dr Martin as a co-author for her integral part in the acoustic measurements.

Reviewer 1

This study used a theta burst transcranial ultrasound stimulation (TUS) protocol to stimulate the posterior cingulate (PCC) and dorsal anterior cingulate cortex (dACC) in 24 healthy subjects. It was found that functional connectivity measured by fMRI increased in both stimulated regions but mostly at a later time point of ~46 min rather than an earlier time point of ~ 13 min after TUS. GABA levels measured by MR spectroscopy was reduced in the PCC and not the dACC after TUS. The authors concluded that the effects of TUS are state dependent.

This is an interesting and novel study on the effect of theta burst TUS. There are limitations to the study such as the size of the ultrasound field and the overlap with MRS voxel was different for the two targets. There are other issues that need to be considered.

We thank the reviewer for their positive comments and suggestions to improve our manuscript. The reviewer is right that the size of the -6 dB ultrasound field in the PCC and dACC were significantly different and therefore their overlap with MRS voxel were different too. This is an interesting point and one that we have spent some time considering too. First, it is important to note that the described field sizes and boundaries are not equivalent to the area of the biological effect. We believe that TUS neuromodulation has some form of dose dependence, and that the field strength should be described as a function of that dependence. In this light, tissue at the focal peak will be dosed twice that of tissue along the FWHM boundary. However, we did test whether the focal volume (the volume within the FWHM boundary) was related to the effects measured in the study (network and neurochemistry changes). We did not observe any significant relationships and thus cannot argue that this was driving the differential effects observed between the regions.

Major Comments

1. The conclusion that the effect of TUS is state dependent is not convincing. The main argument is that the PCC is active during wakeful rest. However, another (perhaps more likely) possibility is that the effects are location dependent. Possibilities other than state dependency was not mentioned in the Abstract or the Introduction.

We agree that we did not convincingly show that state dependency was the main driver of the differences in TUS effects between regions. While we did discuss other possible effects like location and cell-type dependency in our Discussion section, we had placed too much emphasis on state dependence in the Abstract and Introduction sections. We have revised the Abstract and Introduction to reflect this, as follows:

Changes made to Abstract (pg. 2): “The difference in TUS effects on the posterior and anterior cingulate could suggest state- or location-dependency of the TUS effect – both mechanisms increasingly recognized to influence the brain’s response to neuromodulation.”

Changes made to Introduction (pg. 5): “The disparate findings between PCC and dACC stimulation suggest a possible state- or location-dependence of TUS, with important implications for the design and development of future TUS research in humans.”

2. Sound was played to the subjects only in the sham condition. How the sound was set to be similar to the auditory effects of TUS was not clear. This makes the conditions different for the real versus sham TUS and could potentially account for some of the results. It is not clear why the sound was not played in the real TUS conditions.

We have now described how we designed the sound to be similar to the auditory effects of TUS and added the following to the methods section (pg. 23): “We designed the waveform to match the ultrasound pulses produced by the TUS protocol (frequency, pulse repetition frequency and pulse duration). We adjusted the sampling rate to change the pitch of the sound and asked five lab members who have experienced the TUS protocol to vote on the sound that best matched the TUS and played this during the sham condition.”

We did not play the sound during the real TUS conditions because it would have been impossible to perfectly overlap the timings of the two sounds – i.e., participants would be able to discern two separate sounds during the real TUS conditions and only one sound during the sham condition, which we think would contribute even more to a potential difference between the real TUS and sham conditions. Sounds that are played at the same time can either have a constructive or destructive effect based on whether they are in or out of phase with each other. However, the phase of the bone conducted sound cannot be perfectly quantified in time as it is dependent on individual skull shape or properties.

We have now included an explanation for this in the methods section (pg. 23): “Headphones were placed on the participant’s head ..., and the sound was only played during the sham session so that participants do not hear two separate sounds (i.e., the actual sound from the TUS protocol and the audio played through the headphones) during verum TUS conditions.”

3. The subjects were obviously not blinded to the study conditions. I assume that the experimenters and the investigators who analyzed the data was not blinded. This should be stated and acknowledged as a limitation of the study.

The subjects were not blinded to the **location of TUS** because of where we needed to place the transducer to target each location, i.e., over the anterior or posterior cingulate for the two verum TUS conditions, or over the mid-cingulate during the sham condition. However, we believe that participants were blinded to the **TUS study conditions** in that they were not aware that positioning the TUS over the mid-cingulate was for the sham condition. Participants were asked at the end of the study whether they were aware that we did not apply TUS during one of the sessions. All participants, except those who were part of the lab and had experienced TUS before and were aware that there was a sham condition (n = 4), reported that they were unaware that there was a sham condition and, importantly, could not identify which session was the sham session. We have now included a short paragraph to clarify this in the discussion (pg. 20): “We tried to design the sound delivered via bone conduction headphones during the sham condition to closely match the TUS condition. A small number of participants (n = 4) who were familiar with the TUS equipment were able to identify the sham condition, however, the majority participants (n = 20) were unaware that we had a sham condition when asked after completing the study. Thus, we believe that the majority of participants were sufficiently blinded to the study conditions.”

The experimenters and investigators who analyzed the data were not blinded and we have acknowledged this as a limitation of the study (pg. 20 of discussion): “Another possible limitation is that the study was designed as a single-blind study: the experimenters and the researchers who analysed the data were aware of the study conditions, which could possibly introduce some bias during the data collection and analysis.”

4. In the fMRI analyses, the first and second runs were compared to the average of the two runs in the sham session. However, there could be non-specific effects of time in the study. The first run in the real TUS conditions should be compared to the first run of the sham condition, and the second run in the real TUS conditions compared to the second run of the sham condition.

We had compared the first and second sham runs to each other and did not find any significant differences (see Supplementary Information). We also ran the comparisons between real TUS and sham conditions by run as suggested (i.e. first run of TUS compared with first run of sham and second run of TUS with second run of sham) and the pattern of results were similar to the pattern when the mean of the two runs was used, but these differences were not statistically significant. This is why we decided to combine the sham runs to increase the power to identify the differences in connectivity. We have now included these additional analyses in the Supplementary Material and added the following changes to the manuscript to clarify why we had combined the two runs in the sham condition (pg. 28-29): “Comparisons of each TUS session against the corresponding run in the sham condition (i.e. run 1 TUS vs run 1 sham and run 2 TUS vs run 2 sham) yielded results which were similar in spatial extent but not statistically significant (Supplementary Figure 4). Since there were no statistically significant differences between the two runs in the sham condition, we combined the sham runs to increase the fMRI signal to noise ratio and our power to detect differences between active TUS and sham conditions.”

Minor comments

5. Based on the Figure 3 and the coordinates provided, the left PCC and dACC were targeted but this was not stated in the manuscript.

Thank you - this has now been corrected in the manuscript throughout.

6. Pg. 18 What “the regime of the cells” mean is not clear.

Thank you - we have now made this clearer on pg. 18: “firing regime (e.g. high frequency, irregular/infrequent or absent firing)”. We had meant the firing regime or firing frequency of the underlying cells.

7. Pg. 20. It is not clear what categories of mediations were not allowed.

We have now made it clear that we exclude pro-convulsive drugs (or participants who are in withdrawal from these drugs). Changes to pg. 21: “... prescription or non-prescription medication (pro-convulsive drugs, drugs (or combinations of drugs) that lower seizure threshold or withdrawal from drugs that incurs a lowering of seizure threshold (i.e. anti-convulsant withdrawal))”.

8. A previous study (PMID: 36228977) has shown that there are wide spread changes in brain oscillations and connectivity with theta burst TUS to the motor cortex. This should be discussed in relation to the current findings.

Thank you for bringing our attention to this study. We have included this in our discussion on pg. 17: “Using TMS, magnetoencephalography (MEG) and the same theta-burst TUS protocol, Samuel and colleagues (Samuel et al., 2022) reported increased motor cortex excitability as well as increased local MEG coherence in motor regions, with TMS changes being correlated with the MEG coherence measures.”

Reviewer 2

The authors used the new technique of transcranial ultrasound sonification (TUS) to stimulate deep cortical areas dACC and PCC and compared this stimulation with a sham stimulation. dACC is the crucial area of the salience network and PC is the crucial area of the default mode network. The stimulation protocol consisted of a theta-burst stimulation for 80 seconds. The simulation of the stimulation effects confirmed focal stimulation with an increase of temperature at the stimulation sites below 2° C. The measurement of GABA concentration at the sites of stimulation revealed that the GABA concentration decreased in PCC, but not in dACC. The stimulation of dACC increased the activity of the salience network and the stimulation of PCC increased the activity of the default mode network. The ROI based connectivity analysis resulted also in increased of connectivity of both areas. The effects of the TUS on neuroplasticity lasted for at least 50 minutes and was apparently state dependent.

The study includes the novel technique of TUS and sheds new light on the effects of this non-invasive

stimulation technique. The study design is elaborate and sound. The involved brain imaging techniques are high standard. The methods of data analysis are elaborated and the result are plausible and very interesting.

We thank the reviewer for their encouraging comments.

I have only two questions:

1. A question concerning the theta-burst stimulation protocol. The stimulation consisted of 20 ms pulses, spaced by 200 ms pause. It seems to me that the authors used a continuous theta-burst protocol. It is known from the transcranial magnetic stimulation that continuous theta-burst stimulation is inhibitory and intermittent theta-burst stimulation is excitatory. How can the authors be sure that μ protocol they used was excitatory?

Thank you for this interesting comment. The reviewer is correct that this approach is similar to continuous theta burst TMS protocol (cTBS). However, cTBS consists of 3 TMS pulses delivered at 50Hz, with a 5Hz interburst interval. The total number of pulses delivered is 600. The TUS applied here is rather more similar to standard 5Hz repetitive rTMS, albeit for a much shorter period than would normally be used. The cortical excitability effects of 5Hz TMS are somewhat inconsistent across papers, but it is broadly thought to be excitatory (Fitzgerald et al., 2006), particularly for longer pulse durations (for TMS this has been assessed for pulse lengths up to 200 μ s (Halawa et al., 2021)). While it always difficult to extrapolate between brain stimulation approaches, especially when applied to different brain regions, it would seem at least plausible therefore that our TUS protocol is likely to be excitatory.

However, we did not conclusively state that the protocol we used was excitatory. Previous studies using the same protocol have shown increased motor cortex excitability (Zeng et al., 2021) and increase in local MEG connectivity in motor areas (Samuel et al., 2022) when theta-burst TUS was applied to the motor cortex. Our results complement these findings and show that there was a change in excitability in the regions we targeted: 1) reduced GABA in the PCC, which could suggest either disinhibition (e.g., silencing of GABAergic interneurons), increased excitation (e.g., increased glutamatergic activity leading to a downregulation of conversion of glutamate to GABA) or both, and 2) increased functional connectivity, as evidenced by the fMRI findings in both PCC and dACC. These findings suggest increased excitability, although this is still uncertain and further studies (e.g., pharmacological studies) are required to determine the mechanism. The disparate findings in the PCC and dACC may also suggest that the effect of a specific TUS protocol, like with other brain stimulation techniques (Castrillon et al., 2020), may not strictly be excitatory or inhibitory, but may depend on the underlying state and composition of the brain region.

2. The GABA concentration decreased significantly in PCC, but not in dACC. For the assessment of the proton magnetic resonance spectroscopy (MRS) ROIs of 2 x 2 cm, which is rather low for a 3T MRS. Would the significance increase if the authors used for example 3 x 3 cm voxels?

We agree that 2 x 2 x 2 cm³ (i.e. 8 cm³) voxels are on the lower end of the MRS voxel size range at 3T MRI (typically 3 x 3 x 3 cm³ or 27 cm³), and larger voxel sizes would indeed increase the SNR. However, the TUS focus is small and therefore by increasing the volume of the MRS voxel, we would potentially be diluting the measured change in GABA due to TUS. Although it is still uncertain how the TUS focus size relates to the area of the biological effect, we believe that TUS neuromodulation

has some form of dose dependence on intensity and volume. Tissue at the TUS focal peak will be dosed twice that of tissue along the FWHM boundary. Thus, we wanted our MRS measurement volume to be as close to the TUS focus size as possible to be able to capture any changes in GABA undiluted by tissue outside the volume affected by TUS. Our pilot data suggested that $2 \times 2 \times 2 \text{ cm}^3$ was the smallest voxel size achievable that would produce an acceptable SNR (given the constraints imposed by length of scan (signal averages) and field strength).

In 4 pilot subjects, we acquired $3 \times 3 \times 3 \text{ cm}^3$ voxels, $2 \times 2 \times 2 \text{ cm}^3$ voxels and $1.5 \times 1.5 \times 2 \text{ cm}^3$ voxels with varying numbers of averages (200, 150, 128). The spectra looked qualitatively good up to $2 \times 2 \times 2 \text{ cm}^3$ but there was very obviously much higher noise with the smallest voxel size. The GABA+/water model fit error at $2 \times 2 \times 2 \text{ cm}^3$ was $\sim 5.05\text{-}6.7\%$, while the fit error at $3 \times 3 \times 3 \text{ cm}^3$ was $\sim 4.5\%$ for the same number of averages. The fit error was much higher ($\sim 9\text{-}17\%$) at the lower voxel size even with more averages, as expected. We decided that we would accept the slightly higher fit error with $2 \times 2 \times 2 \text{ cm}^3$ voxels so that the MRS measurement volume was closer to that of the TUS focus size.

Several other studies have reported good spectra at similar voxel sizes at 3T ((Stagg et al., 2009) – 8 cm^3 voxels, (Iwabuchi et al., 2017)– $8.1\text{-}9 \text{ cm}^3$ voxels, (Nandi et al., 2022) – 8 cm^3 voxels at both 7T and 3T). In their methodological consensus and recommendations paper, Wilson and colleagues (Wilson et al., 2019) suggested increasing the number of averages when acquiring MRS at lower MR field strength or smaller voxel sizes. Their recommendation for $2 \times 2 \times 2 \text{ cm}^3$ voxels at 3T MRI was 64 averages. We acquired our data with 128 averages. From the quality checks on our data (FWHM, GABA+/water SNR, and model fit errors), we believe the data we collected is of high quality. Unfortunately, we do not have the data to retrospectively determine whether we would see a significant increase in the dACC if we had used larger MRS voxels, but given that the TUS effect would probably be diluted by measuring over a larger volume, we think this would not be the case.

References:

- Castrillon, G., Sollmann, N., Kurcyus, K., Razi, A., Krieg, S. M., & Riedl, V. (2020). The physiological effects of noninvasive brain stimulation fundamentally differ across the human cortex. *Science Advances*, 6(5). <https://doi.org/10.1126/sciadv.aay2739>
- Fitzgerald, P. B., Fountain, S., & Daskalakis, Z. J. (2006). A comprehensive review of the effects of rTMS on motor cortical excitability and inhibition. *Clinical Neurophysiology: Official Journal of the International Federation of Clinical Neurophysiology*, 117(12), 2584–2596. <https://doi.org/10.1016/j.clinph.2006.06.712>
- Halawa, I., Reichert, K., Aberra, A. S., Sommer, M., Peterchev, A. V., & Paulus, W. (2021). Effect of Pulse Duration and Direction on Plasticity Induced by 5 Hz Repetitive Transcranial Magnetic Stimulation in Correlation With Neuronal Depolarization. *Frontiers in Neuroscience*, 15(November), 1–10. <https://doi.org/10.3389/fnins.2021.773792>
- Iwabuchi, S. J., Raschke, F., Auer, D. P., Liddle, P. F., Lankappa, S. T., & Palaniyappan, L. (2017). Targeted transcranial theta-burst stimulation alters fronto-insular network and prefrontal GABA. *NeuroImage*, 146, 395–403. <https://doi.org/10.1016/j.neuroimage.2016.09.043>
- Nandi, T., Puonti, O., Clarke, W. T., Nettekoven, C., Barron, H. C., Kolasinski, J., Hanayik, T., Hinson, E. L., Berrington, A., Bachtiar, V., Johnstone, A., Winkler, A. M., Thielscher, A., Johansen-Berg, H., & Stagg, C. J. (2022). tDCS induced GABA change is associated with the simulated electric field in M1, an effect mediated by grey matter volume in the MRS voxel. *Brain Stimulation*, 15(5), 1153–1162. <https://doi.org/10.1016/j.brs.2022.07.049>
- Samuel, N., Zeng, K., Harmsen, I. E., Ding, M. Y. R., Darmani, G., Sarica, C., Santyr, B., Vetkas, A., Pancholi, A., Fomenko, A., Milano, V., Yamamoto, K., Saha, U., Wennberg, R., Rowland, N. C., Chen, R., & Lozano, A. M. (2022). Multi-modal investigation of transcranial ultrasound-induced

neuroplasticity of the human motor cortex. *Brain Stimulation*, 15(6), 1337–1347. <https://doi.org/10.1016/j.brs.2022.10.001>

Stagg, C. J., Best, J. G., Stephenson, M. C., O’Shea, J., Wylezinska, M., Kincses, Z. T., Morris, P. G., Matthews, P. M., & Johansen-Berg, H. (2009). Polarity-Sensitive Modulation of Cortical Neurotransmitters by Transcranial Stimulation. *Journal of Neuroscience*, 29(16), 5202–5206. <https://doi.org/10.1523/JNEUROSCI.4432-08.2009>

Wilson, M., Andronesi, O., Barker, P. B., Bartha, R., Bizzi, A., Bolan, P. J., Brindle, K. M., Choi, I. Y., Cudalbu, C., Dydak, U., Emir, U. E., Gonzalez, R. G., Gruber, S., Gruetter, R., Gupta, R. K., Heerschap, A., Henning, A., Hetherington, H. P., Huppi, P. S., ... Howe, F. A. (2019). Methodological consensus on clinical proton MRS of the brain: Review and recommendations. *Magnetic Resonance in Medicine*, 82(2), 527–550. <https://doi.org/10.1002/mrm.27742>

Zeng, K., Darmani, G., Fomenko, A., Xia, X., Tran, S., Nankoo, J., Oghli, Y. S., Wang, Y., Lozano, A. M., & Chen, R. (2021). Induction of Human Motor Cortex Plasticity by Theta Burst Transcranial Ultrasound Stimulation. *Annals of Neurology*, 1–15. <https://doi.org/10.1002/ana.26294>

Reviewer #1 (Remarks to the Author):

The authors addressed most of the comments from the previous review. I have one minor point that I believe is worth clarifying. The authors now indicate that subjects who use "pro-convulsive" drugs or who were withdrawing from anticonvulsants were excluded. I am still not clear what "pro-convulsive" drugs were not allowed. It would be helpful to give examples or categories of drugs were not allowed (e.g. tricyclic antidepressants) and what were allowed, and how it was determined if a drug is "pro-convulsive" or not. I am not clear if common drugs such as SSRI or SNRI were allowed. Moreover, it would appear that benzodiazepines were allowed since they are not "pro-convulsive" but they affect the CNS. If this is the case, it would be helpful to indicate how many subjects were on CNS active drugs (e.g. X subjects were on benzodiazepines, Y subjects were on SSRI etc). Did the authors ask about use of recreational drugs?

Reviewer #2 (Remarks to the Author):

The authors have answered my questions sufficiently.

REVIEWERS' COMMENTS

Reviewer #1 (Remarks to the Author):

The authors addressed most of the comments from the previous review. I have one minor point that I believe is worth clarifying. The authors now indicate that subjects who use “pro-convulsive” drugs or who were withdrawing from anticonvulsants were excluded. I am still not clear what “pro-convulsive” drugs were not allowed. It would be helpful to give examples or categories of drugs were not allowed (e.g. tricyclic antidepressants) and what were allowed, and how it was determined if a drug is “pro-convulsive” or not. I am not clear if common drugs such as SSRI or SNRI were allowed. Moreover, it would appear that benzodiazepines were allowed since they are not “pro-convulsive” but they affect the CNS. If this is the case, it would be helpful to indicate how many subjects were on CNS active drugs (e.g. X subjects were on benzodiazepines, Y subjects were on SSRI etc). Did the authors ask about use of recreational drugs?

We have now provided a full list of drugs that meet our exclusion criteria. No subjects were on CNS active drugs and this has now been clarified in the main text. We did not specifically ask about use of recreational drugs, but we did ask participants to disclose **any** prescription or non-prescription medication or supplements they are currently taking or have recently stopped taking. None of the participants disclosed taking any recreational drugs at the time of the study.

Reviewer #2 (Remarks to the Author):

The authors have answered my questions sufficiently.

Thank you.